# Insulin-like Growth Factor-I Reduces Collagen Production by Hepatic Stellate Cells Through Stimulation of Collagen Degradation System via mTOR-Dependent Signaling Pathway

**DOI:** 10.3390/biomedicines13030566

**Published:** 2025-02-24

**Authors:** Takako Nishikawa, Natsuko Ohtomo, Yukiko Inoue, Mami Takahashi, Hitoshi Ikeda, Kazuhiko Koike, Nobutake Yamamichi, Mitsuhiro Fujishiro, Tomoaki Tomiya

**Affiliations:** 1Department of Gastroenterology, Graduate School of Medicine, The University of Tokyo, Tokyo 113-8655, Japan; natsuko-tky@umin.ac.jp (N.O.); yukiinoue-gi@umin.org (Y.I.); yamamichin-int@h.u-tokyo.ac.jp (N.Y.); fujishiromi-int@h.u-tokyo.ac.jp (M.F.); 2Center for Epidemiology and Preventive Medicine, The University of Tokyo Hospital, Tokyo 113-8655, Japan; mami58727@gmail.com; 3Division for Health Service Promotion, The University of Tokyo, Tokyo 113-0033, Japan; 4Department of Clinical Laboratory Medicine, Graduate School of Medicine, The University of Tokyo, Tokyo 113-8655, Japan; hitoshiikeda.1im@gmail.com; 5Kanto Central Hospital, Tokyo 158-8531, Japan; kkoike-tky@umin.ac.jp; 6Department of Gastroenterology and Hepatology, Saitama Medical University, Saitama 350-0451, Japan; tomiya@saitama-med.ac.jp; 7Health Promotion Center, Saitama Medical University, Saitama 350-0451, Japan

**Keywords:** hepatic stellate cells, IGF-I, matrix metalloproteinase, mTOR, tissue inhibitor of metalloproteinase

## Abstract

**Aim**: The liver is the major source of circulating insulin-like growth factor (IGF)-I. Serum IGF-I levels are decreased in cirrhotic patients depending on severity. IGF-I administration was shown to improve liver function in patients and animal models of liver cirrhosis. However, controversy exists as to whether IGF-I stimulates or reduces fibrosis in the liver. The effects of IGF-I on collagen accumulation by hepatic stellate cells (HSCs) and its mechanisms were studied. **Methods**: A moderately activated HSC clone was used to determine the effect of IGF-I administration on the collagen production system, including its degradation. The intracellular signaling system was also studied in the cells stimulated by IGF-I. **Results**: IGF-I treatment reduced total amounts of collagen deposition in a dose-related manner, while DNA synthesis was stimulated by IGF-I. IGF-I treatment did not affect transforming growth factor-beta levels and type I procollagen mRNA expression. Expression of matrix metalloproteinase (MMP)-2 and -9 was upregulated, and tissue inhibitor of metalloproteinase (TIMP)-1 expression was downregulated by IGF-I treatment. Rapamycin, an inhibitor of mammalian target of rapamycin (mTOR), suppressed phosphorylation of 70 kDa ribosomal protein S6 kinase and eukaryotic initiation factor 4E-binding protein 1, and abrogated IGF-I-induced increase in MMP-2 and -9 expression and decrease in TIMP-1 expression. **Conclusions**: IGF-I has the ability to stimulate the collagen degradation system by HSCs through an mTOR-dependent pathway independent of modulation of the activation state of HSCs.

## 1. Introduction

Insulin-like growth factor (IGF)-I is a potent polypeptide that has a wide range of biological activities, such as stimulation of growth, survival, and function in various types of cells. Circulating IGF-I is predominantly produced by the liver, mainly hepatocytes [1]. Serum IGF-I levels are decreased in patients with liver cirrhosis depending on the degree of liver dysfunction, and the decrease is associated with poor prognosis [1,2]. Several reports demonstrated beneficial effects of IGF-I supplementation in animal models and in patients complicated with liver cirrhosis. IGF-I administration promoted weight gain, nitrogen retention, and intestinal absorption of nutrients in rats with fibrotic liver [3]. In addition, IGF-I exerted hepatoprotective effects and improved liver function in cirrhotic rats [4,5]. A pilot study of IGF-I supplementation therapy attenuated liver dysfunction in patients with liver cirrhosis [6]. These findings indicate the possibility that IGF-I is used for the treatment of liver cirrhosis. However, the effect of IGF-I on fibrosis itself in fibrotic liver has not been defined.

Hepatic fibrosis results from excess deposition of extracellular matrix, especially type I collagen. Collagens in the liver are considered to be produced mainly by hepatic stellate cells (HSCs), which, when activated, transform into myofibroblast-like cells, proliferation is accelerated, collagen synthesis is increased, and the production of transforming growth factor (TGF)-beta, a potent cytokine that stimulates collagen synthesis, is upregulated. On the other hand, collagen deposition is controlled by its degradation system consisting of matrix metalloproteinases (MMPs) and tissue inhibitors of metalloproteinase (TIMPs); MMPs induce collagen degradation, while TIMPs antagonize MMP activity.

### Effects of IGF-I on Hepatic Fibrogenesis Are Controversial

Previous reports showed that IGF-I stimulated the proliferation of human and rat hepatic stellate cells in primary culture, as well as fibroblasts in culture, and induced collagen synthesis [7,8]. IGF-I was postulated to play an important role in the pathogenesis of fibrosis. On the contrary, IGF-I has been reported to exert antifibrogenic effects in cirrhotic rats [4]. IGF-I administration improved fibrosis histologically and reduced hepatic hydroxyproline content, prolyl hydroxylase activity, and the expression of collagen mRNA, alpha-smooth muscle actin, and TGF-beta. In addition, IGF-I upregulated MMPs with decreased expression of TIMPs in the cirrhotic liver of rat models [5]. When a hepatic stellate cell clone (cHSC) was incubated with conditioned media of IGF-I-producing cells, activation of HSCs was suppressed [9]. The reasons for these diverse effects of IGF-I on HSCs and collagen production are uncertain. HSCs are quite heterogeneous regarding their activation status and phenotypes, such as the expression of collagen mRNAs, TGF-beta, MMPs, and TIMPs in fibrotic liver. It is also known that HSCs in primary culture are activated during culture, and IGF-I receptor expression is diminished. Experimental conditions and characteristics of HSCs used for the studies might influence the outcomes. In addition, cells other than HSCs stimulated by IGF-I might contribute to the overall effect of IGF-I on fibrogenesis and fibrolysis in vivo. To elucidate the direct effects of IGF-I on hepatic stellate cells along with collagen production, further point-by-point studies are required.

We studied collagen production and expression of related factors utilizing a cHSC, which maintains the phenotype of moderately activated hepatic stellate cells during culture, including the expression of IGF-I receptor, to elucidate the role of hepatic stellate cells on hepatic fibrogenesis when IGF-I is administered to cirrhotic livers.

## 2. Materials and Methods

### 2.1. Cell Culture

A hepatic stellate cell clone (cHSC), CFSC-8B, was kindly provided by Dr. Marcos Rojkind [10,11]. Cells were plated on plastic culture dishes at 4 × 10^4^ cells/cm^2^ and cultured, as previously described, in Eagle’s minimal essential medium (MEM, Nissui Pharmaceutical Co., Ltd., Tokyo, Japan) containing 10% (*v*/*v*) fetal calf serum (FCS, Gibco Laboratories, Life Technologies Inc., Grand Island, NY, USA). After a 4 h attachment period, the culture medium was replaced with MEM supplemented with 0.5% FCS [12,13]. Cells were used for the following experiments 20 h after plating.

### 2.2. Experiments

Experiment I: The medium was changed to Hank’s balanced salt solution (HBSS) containing various concentrations of recombinant human IGF-I (Wako Pure Chemical Industries, Ltd., Osaka, Japan) and 0.1 mmol/L of 5-bromo-2′-deoxy-uridine (BrdU). Twenty-four hours later, the cells were harvested for the determination of BrdU incorporation into cellular DNA. The number of cells and collagen deposition in the culture well were determined for both 24 and 48 hours after IGF-I addition.

Experiment II: The medium was changed to HBSS with increasing concentrations of IGF-I in the medium. Type I collagen mRNA expression of cHSC and TGF-beta 1 levels in the medium were determined 48 h after IGF-I addition.

Experiment III: A total of 100 ng/mL of IGF-I was added to the medium with or without 30 min pre-incubation of cHSC with 0.1–2.5 nM of rapamycin. After 10- and 40-min incubation periods, Akt and extracellular signal-regulated kinase (ERK) 1/2 phosphorylation and 70 kDa ribosomal protein S6 (p70 S6) kinase and eukaryotic initiation factor 4E-binding protein (4E-BP) 1 phosphorylation, respectively, were evaluated by Western blotting. ProMMP-2 and proMMP-9 activities in the medium and cellular levels of MMP-9 and TIMP-1 were determined 48 h after IGF-I addition.

All study protocols conformed to and were approved by the guidelines of the Faculty of Medicine, University of Tokyo.

### 2.3. Assays

#### 2.3.1. Determination of DNA Synthesis, Number of Cells, and Collagen Deposition of cHSC

DNA synthesis was determined by the detection of BrdU incorporated into cellular DNA during 24 h incubation period by enzyme-linked immunosorbent assay (ELISA) using a commercial kit (Cell proliferation ELISA, BrdU, Roche Molecular Biochemicals, Mannheim, Germany). Number of cells was counted by water-soluble tetrazolium salt-8 assay (Cell counting kit-8, Dojindo Laboratories, Co., Ltd., Kumamoto, Japan). Collagen deposition was assessed by the sirius red dye binding method (Sircol collagen assay kit, Biocolor Ltd., Belfast, UK), according to the manufacturer’s instructions. The total cellular protein was measured by Bradford’s method.

#### 2.3.2. Quantitative Reverse-Transcription Polymerase Chain Reaction (RT-PCR) for Type I Alfa 1 Procollagen mRNA

Total RNA was isolated from cHSC using TRizol (Invitrogen, Carlsbad, CA, USA) according to the manufacturer’s instructions. One microgram of total RNA was reverse-transcribed with the ReverTra Ace^R^ qPCR RT Master Mix for real-time PCR (Toyobo Co., Ltd., Osaka, Japan). Quantitative real-time PCR for rat type I alfa 1 procollagen was performed using the THUNDERBIRD^R^ Probe qPCR Mix (Toyobo Co., Ltd., Osaka, Japan). The reaction was performed in 2 µL cDNA for each analyzed sample using these reagents. The primers and probe used were as follows. Sense: 5′-TTCACCTACAGCACGCTTGTG-3′, antisense: 5′-GATGACTGTCTTGCCCCAAGTT-3′, and probe: 5′-ATGGCTGCACGAGTCACACCG-3′ [14]. Eukaryotic 18S primers and probe were obtained from Applied Biosystems (Hs99999901-S1, TaqMan Gene Expression Assays, Foster City, CA, USA).

Samples were incubated for 10 min at 95 °C, followed by 40 cycles at 95 °C for 15 s and 60 °C for 1 min. The target gene mRNA expression level was relatively quantified to ribosomal 18S using the 2^−ΔΔCT^ method (User Bulletin #2, Applied Biosystems, Foster City, CA, USA).

#### 2.3.3. Determination of TGF-Beta 1 Levels and MMPs Activity in the Medium

Activated TGF-beta 1 levels in the culture medium of cHSC were determined using a rat TGF-beta 1 ELISA kit (R&D Systems, Inc., Minneapolis, MN, USA). MMP activities were detected by a zymography kit (COSMO BIO, Co., Ltd., Tokyo, Japan) [15,16].

#### 2.3.4. Determination of Expression of MMP-9, TIMP-I, Akt, ERK1/2, p70 S6 Kinase, and 4E-BP1 by Western Blotting

Expression of MMP-9 and TIMP-1 proteins was determined by Western blotting using an anti-MMP-9 antibody (sc-10737, Santa Cruz Biotechnology, Santa Cruz, CA, USA) and an anti-TIMP-1 antibody (sc-6834, Santa Cruz Biotechnology, Santa Cruz, CA, USA), respectively, as described previously [17]. Total and phospho-p70 S6 kinases were detected using anti-p70 S6 kinase antibody (#9202 Cell Signalling Technology, Inc., Danvers, MA, USA) and anti-phospho-p70 S6 kinase antibody (#9205 Cell Signalling Technology, Inc., Danvers, MA, USA), respectively [12]. Electrophoretic mobility was studied using anti-4E-BP1 antibody (sc-6024 Santa Cruz Biotechnology, Santa Cruz, CA, USA) [12,13]. Total and phospho-Akt and total and phospho-ERK1/2 were detected using anti-Akt antibody (#4691 Cell Signalling Technology, Inc., Danvers, MA, USA), anti-phospho-Akt antibody (#4060 Cell Signalling Technology, Inc., Danvers, MA, USA), anti-ERK1/2 antibody (#4695 Cell Signalling Technology, Inc., Danvers, MA, USA) and anti-phospho-ERK1/2 antibody (#4370 Cell Signalling Technology, Inc., Danvers, MA, USA), respectively.

In brief, cells were lysed in ice-cold buffer A (50 mM Tris–HCl, pH 8.0, 1% Nonidet P-40, 120 mM NaCl, 20 mM NaF, 1 mM EDTA, 6 mM EGTA, 20 mM β-glycerophosphate, 0.5 mM dithiothreitol, 50 μM p-APMSF, 1 μg/mL aprotinin, and 1 μg/mL leupeptin) and incubated for 10 min. They were centrifuged at 10,000× *g* for 5 min at 4 °C. An aliquot of the supernatants was mixed with a one-fifth volume of SDS buffer (10% SDS, 30% glycerol, 280 mM Tris–HCl, pH 6.8, and 0.6 M dithiothreitol), separated by SDS–PAGE on a 7.5% gel, and transferred to a PVDF membrane. Phospho- and total-kinases were detected by Western blotting using anti-phospho- and anti-total-antibodies. Electophoretic mobility was studied using an anti-4E-BP1 antibody, as described previously.

MMP9 was examined using a 7.5% gel, and TIMP1 and 4EBP1 were examined using a 15% gel.

### 2.4. Statistical Analyses

The differences between two unpaired samples were defined as significant when *p*-values by Student’s *t* test were less than 0.05.

## 3. Results

### 3.1. Proliferation of cHSC Stimulated by IGF-I

Twenty-four-hour incubation with IGF-I enhanced BrdU incorporation by cHSC in a dose-related manner (Figure 1A) without significant changes in the number of cells (Figure 1B). Dose-related increase in the number of cells was observed when the incubation period was 48 h (Figure 1C).

### 3.2. Effect of IGF-I Treatment on Collagen Deposition, Type I Alfa 1 Procollagen mRNA Expression, and TGF-Beta 1 Levels in the Culture Medium of cHSC

Collagen deposition in the culture well was reduced for both 24 and 48 h after IGF-I addition (Figure 2A,B). However, as shown in Figure 2C, type I alfa 1 procollagen mRNA levels were not affected by IGF-I treatment for 48 h. TGF-beta 1 levels in the culture medium of cHSC showed no significant changes by IGF-I treatment for 48 h (Figure 2D).

### 3.3. Expression of MMPs and TIMP-1 by cHSC in the Presence of IGF-I

Incubation with IGF-I for 48 h enhanced proMMP-2 and proMMP-9 activities and MMP-9 protein expression and suppressed TIMP-1 protein expression in cHSC (Figure 3A–C).

### 3.4. Activation of Akt, ERK1/2, p70 S6 Kinase, and 4E-BP1 in cHSC Stimulated by IGF-I and Inhibitory Effect of Rapamycin on the Activation

IGF-I upregulated phosphorylation of Akt, ERK1/2, p70 S6 kinase, and 4E-BP1 in cHSC (Figure 4A).

Phosphorylation of Akt1 and ERK1/2 was upregulated in a dose-dependent manner of IGF-1 addition. P70s6 kinase was upregulated by 500 ng/mL IGF-I administration. Phosphorylation of 4EBP-1 was also upregulated in a dose-dependent manner. In 4EBP-1, the beta band, which indicates a moderate phosphorylated form, was upregulated up to 100 ng/mL of IGF-I administration. With 500 ng/mL IGF-I administration, the gamma band, which has an even higher degree of phosphorylation than the beta band, was enhanced.

Rapamycin suppressed phosphorylation of p70 S6 kinase and 4E-BP1 but not Akt and ERK1/2 in cHSC (Figure 4B).

Phospho-p70 S6 kinase was downregulated at 2.5 nM of rapamycin administration. γ band was also downregulated at 2.5 nM of rapamycin administration.

### 3.5. Effect of Rapamycin on Expression of MMPs and TIMP-1 by cHSC in the Presence of IGF-I

Rapamycin suppressed proMMP-2 and proMMP-9 activities and MMP-9 protein expression stimulated by IGF-I in cHSC. In addition, rapamycin canceled the inhibitory effect of IGF-I on TIMP-1 expression (Figure 5A–D).

## 4. Discussion

In the present study, we showed that IGF-I reduced collagen production by HSCs through the activation of the degradation system of collagen.

Freshly isolated HSCs show drastic changes in phenotypes during culture, including a rapid decline in IGF-I receptor expression, while cloned HSCs showed a variety set of phenotypes depending on the characters of originated HSCs. For example, some cHSCs lack the ability to produce MMPs [18]. We utilized a cHSC, which has a small amount of fat droplets containing vitamin A [10,11] and is postulated to be moderately activated. This cell line has the ability to produce a set of factors related to collagen synthesis and degradation, including collagens themselves, TGF-beta, MMPs, and TIMP-1. In addition, it expresses the IGF-I receptor continuously.

By using moderately activated hepatic stellate cells rather than quiescent hepatic stellate cells, we were able to study the dynamics of stellate cells when IGF-I is administered to cirrhotic livers. IGF-I has been reported to possess the ability to stimulate the proliferation of various kinds of cells [1]. This finding implies that IGF-I is one of the activators of HSCs since accelerated proliferation is one of the features of activated HSCs.

Along with proliferative activity, mRNA of type 1 collagen, the principal collagen of the fibrotic liver, was also shown to be upregulated by IGF-I in primary cultured HSCs, supporting the notion that IGF-I is an activator of HSCs [7]. However, Sorbrevals et al. recently reported that IGF-I suppressed TGF-beta production, followed by a downregulation of type 1 collagen mRNA in a cHSC, suggesting that IGF-I suppressed collagen synthesis through reduction of TGF-beta stimuli [9], whereas our results indicated that IGF-I did not influence TGF-beta production nor type 1 procollagen mRNA expression in cHSC used in our experiments. No significant effect of IGF-I on TGF-beta production was also reported previously in primary cultured rat HSCs [19]. TGF-beta is considered to be a key molecule involved in HSC activation [20]. TGF-beta synthesis is upregulated during the activation of HSCs, and activated HSCs are the most important source of TGF-beta in fibrogenesis, which is largely driven by TGF-beta. IGF-I seems to not be a “classical” activator of HSCs. The mechanisms of diverse effects of IGF-I on TGF-beta and collagen synthesis in HSCs are uncertain. The ability of HSCs to produce TGF-beta depends on the activation status of HSCs. Furthermore, intracellular signaling of TGF-beta differs between quiescent HSCs and transdifferentiated HSCs, and TGF-beta exerts a stimulatory effect on collagen synthesis only in the latter HSCs [21]. In addition, IGF-I may stimulate quiescent rather than transdifferentiated HSCs more efficiently since IGF-I receptors are abundant in the former cells compared to the latter. HSC activation status might confound the effect of IGF-I stimuli on TGF-beta synthesis and signals and collagen synthesis, resulting in intraexperimental differences.

Homeostasis of the extracellular matrix in the liver is controlled by its synthesis and degradation. In animal experiments, it has been documented that the induction of MMP-9 mutants as TIMP-1 scavengers reduced fibrosis by enhancing collagen resorption, and over-expression of TIMP-1 in the liver delayed regression of fibrosis [22,23]. TIMP-I production has been shown to be upregulated in activated HSCs and enhance subsequent accumulation of matrix. The accumulation is followed by the reactive activation of the degradation system consisting mainly of increased net MMP activity with rapidly declining TIMP-I expression in vivo [24]. IGF-I was reported to enhance the degradation system in the experimental animal models, although mechanisms and cell types that contribute to the effect were not specified [5]. We checked levels of MMP-2 and -9, which are proteases mainly produced by hepatic stellate cells [20] and reported to have the ability to cleave collagen, including type I [25,26], as well as TIMP-1 levels. Our study showed that MMP expression and activity were upregulated, and TIMP-1 expression was downregulated by IGF-I treatment in cHSC. IGF-I seems to directly act on HSCs as an activator of the degradation system without preceding an increase in collagen synthesis.

To confirm this point, we studied intracellular signaling related to the expression of MMP and TIMP. Previous studies on the intracellular signaling of IGF-I reported two primary pathways by which IGF-I signals are transmitted: the PI3-Akt pathway and the ERK pathway. These pathways are shown to be involved in the mediation of various cellular responses to IGF-I stimuli, such as mitogenic, antiapoptotic, and chemotactic activities [27,28,29]. Actually, our results also indicate that both PI3-Akt and ERK pathways in cHSC were activated by IGF-I. In HSCs, cell proliferation and expression of type I collagen have been reported to be regulated by the PI3-Akt signaling pathway and similarly by the ERK-dependent pathway, although the role of the ERK pathway is still controversial [7,30,31]. In eukaryotic cells, one of the most important intracellular signal transducers for protein production is the rapamycin-sensitive pathway associated with the mammalian target of rapamycin (mTOR), which lies downstream of the PI3-Akt signaling pathway. mTOR couples with two factors that are directly linked to protein synthesis: 4E-BP1 and p70 S6 kinase [32,33,34,35]. 4E-BP1 regulates protein synthesis through its association with eukaryotic translation initiation factor 4E. p70 S6 kinase controls the step in the initiation of translation involving the binding of mRNA to the 40S ribosomal subunit [32,33]. The mTOR-dependent pathway has recently been shown to play an important role in the regulation of protein production by HSCs [12,13]. We focused on the mTOR pathway in the study of the modulatory mechanism of MMPs and TIMP-1 productions by IGF-I.

IGF-I rapidly increased p70 S6 kinase activity and phosphorylation of 4E-BP1 in HSCs. The addition of rapamycin, a specific inhibitor of mTOR [34,35,36,37], prevented the upregulation of p70 S6 kinase and 4E-BP1 activities but not Akt and ERK1/2 activities in the presence of IGF-I. Furthermore, rapamycin abrogated IGF-I-induced increases in MMP-2 and MMP-9 expression and decreases in TIMP-1 expression. These results clearly indicate that the mTOR pathway transmits the signals for activation of the collagen degradation system by IGF-I.

## 5. Conclusions

IGF-I possesses the ability to directly stimulate the collagen degradation system consisting of MMPs and TIMP-1 by HSCs through the mTOR-dependent pathway, independent of modulation of the activation state of HSCs and preceding changes of collagen synthesis. This finding supports the IGF-I supplementation therapy for liver cirrhosis in clinical settings, although the effect of IGF-I on net collagen deposits might be influenced by the activation status and related factors of HSCs.

The results of this study indicate the effect of IGF-I in promoting fibrolysis in fibrotic livers and suggest that IGF-I may be useful in treating liver cirrhosis. Further investigation would be required to elucidate the dose of IGF-I supplementation required in vivo without any adverse events.

Further precise in vivo experiments would be required to clarify the clinical relevance of IGF-I administration in the patients.

## Figures and Tables

**Figure 1 biomedicines-13-00566-f001:**
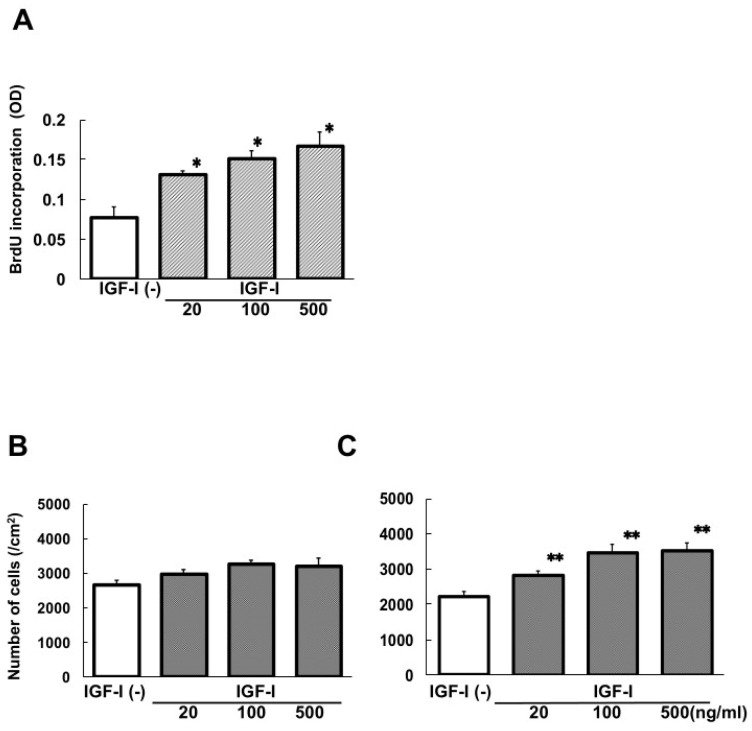
DNA synthesis and number of cells in cHSC cultured in medium supplemented with IGF-I. BrdU incorporation was detected 24 h after the addition of various concentrations of IGF-I (**A**). The number of cells was determined 24 h (**B**) and 48 h (**C**) after incubation with IGF-I. Data were mean ± SEM of eight dishes. * *p* < 0.05 and ** *p* < 0.01 compared to the values without IGF-I addition.

**Figure 2 biomedicines-13-00566-f002:**
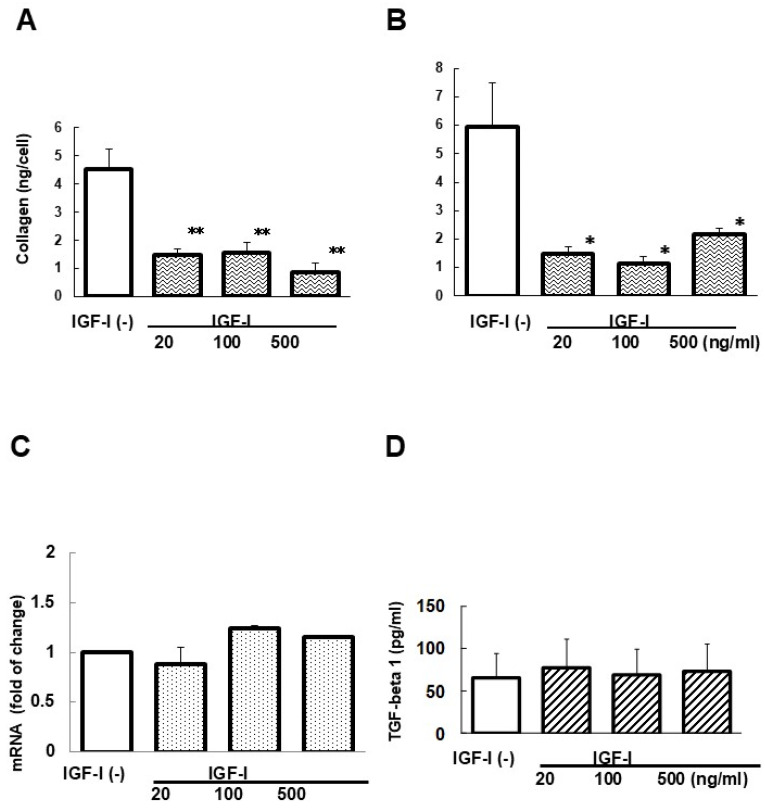
Effect of IGF-I addition on collagen deposition in the culture well, type I alfa 1 procollagen mRNA expression by cHSC, and TGF-beta 1 levels in the medium. Collagen deposition was determined after 24 (**A**) and 48 h (**B**) incubation periods. Data were mean ± SEM of seven dishes. * *p* < 0.05 and ** *p* < 0.01 compared to the values without adding IGF-I. Type I alfa 1 procollagen mRNA was evaluated by RT-PCR after 48 h incubation period (**C**). Data are means ± SEM of three dishes. TGF-beta 1 levels in the medium were determined after 48 h incubation period. Data are means ± SEM of four dishes (**D**).

**Figure 3 biomedicines-13-00566-f003:**
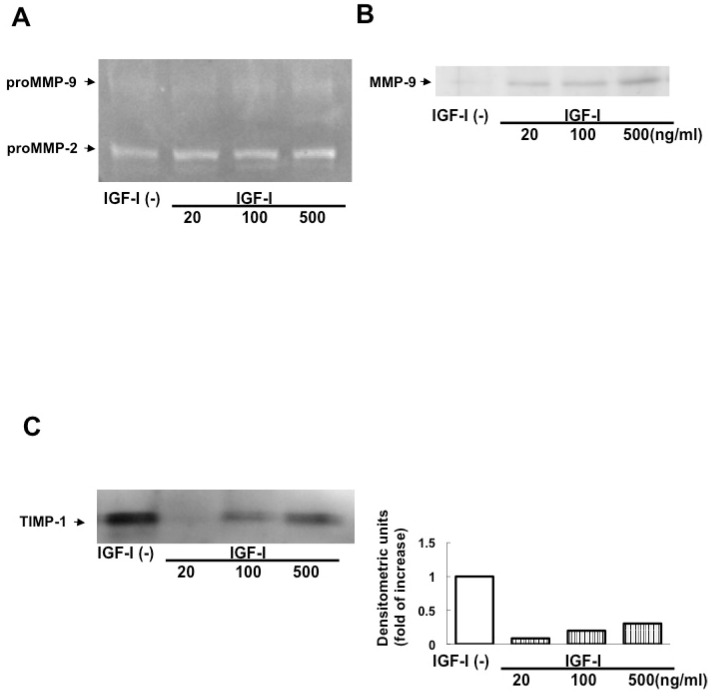
Effect of IGF-I on expression of MMPs and TIMP-1 by cHSC. ProMMP-2 and proMMP-9 activities in cHSC treated by IGF-I for 48 h were determined by zymography (**A**). Expression of MMP-9 and TIMP-1 protein was evaluated by Western blotting analysis (**B**,**C**).

**Figure 4 biomedicines-13-00566-f004:**
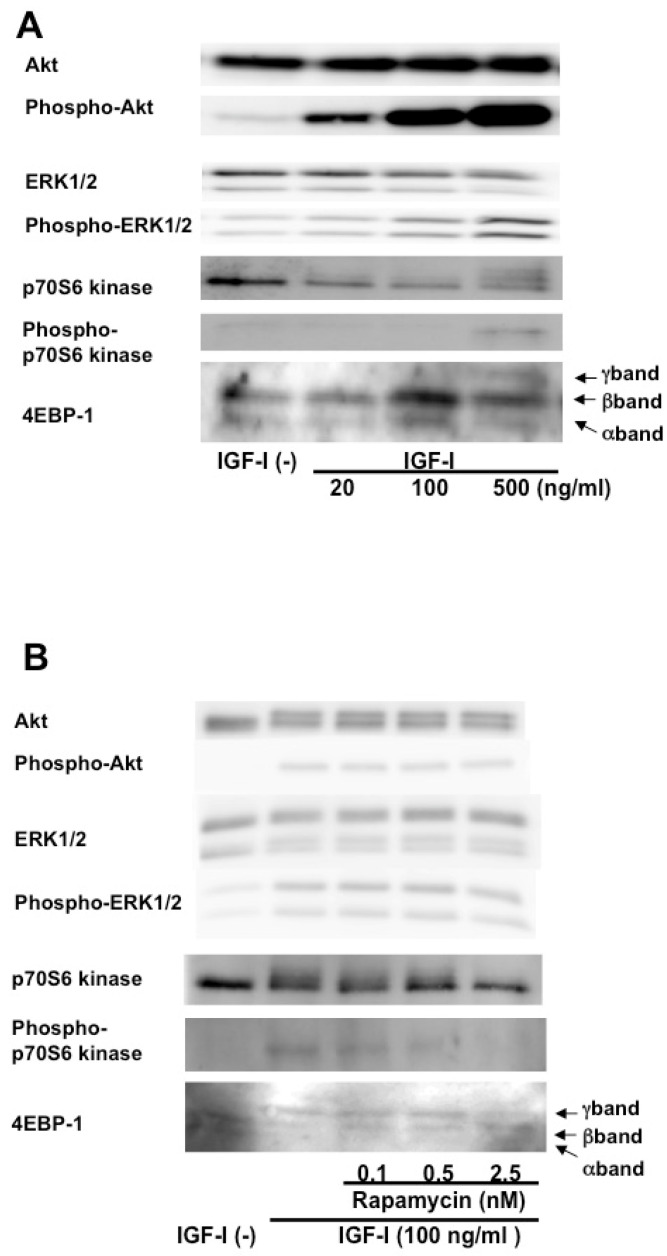
Phosphorylation of Akt, ERK1/2, p70 S6 kinase, and 4E-BP1 in cHSC stimulated by IGF-I and the effect of rapamycin on the phosphorylation. Phosphorylation of Akt, ERK1/2 p70 S6 kinase, and 4E-BP1 (**A**) and the effect of rapamycin on phosphorylation of Akt, ERK1/2, p70 S6 kinase, and 4E-BP1 (**B**) in cHSC cells cultured in medium supplemented with 100 ng/mL of IGF-I were determined. Total and phosphorylated kinase proteins, except for 4E-BP1, were shown in the upper and lower panels, respectively. Faster-migrating bands of 4E-BP1 (α and β bands) indicate unphosphorylated or lower phosphorylated 4E-BP1 and the slower-migrating band (γ band) indicates highly phosphorylated 4E-BP1.

**Figure 5 biomedicines-13-00566-f005:**
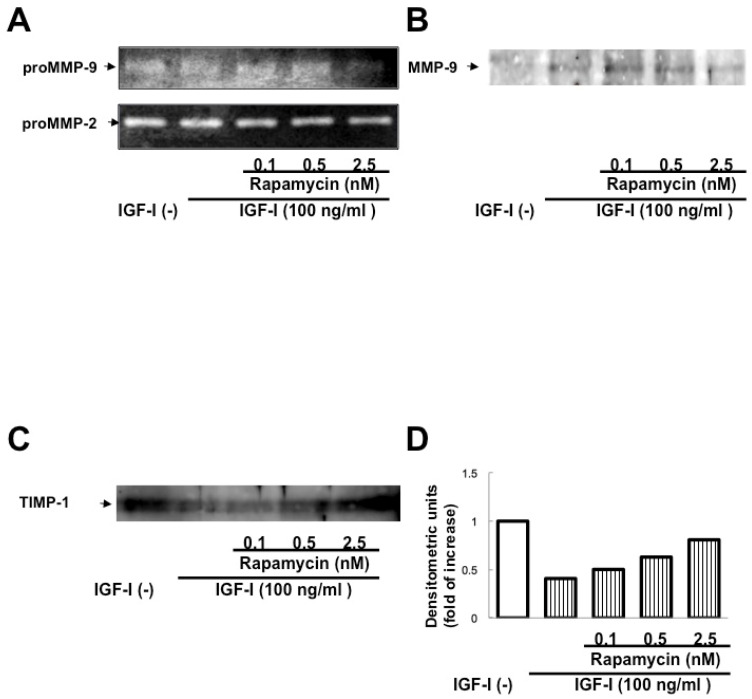
Effect of rapamycin on expression of MMPs and TIMP-1 by cHSC in the presence of IGF-I. Effect of rapamycin on proMMP-2 and proMMP-9 activities in cHSC treated with 100 ng/mL of IGF-I for 48 h was evaluated by zymography (**A**). Expression of MMP-9 and TIMP-1 proteins treated with IGF-I for 48 h in the presence of rapamycin was evaluated by Western blotting (**B**–**D**).

## Data Availability

The data presented in this study are available on request from the corresponding author due to privacy and ethical reasons.

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
