# Peer review of "Insulin-like Growth Factor-I Reduces Collagen Production by Hepatic Stellate Cells Through Stimulation of Collagen Degradation System via mTOR-Dependent Signaling Pathway"

_biomedicines, 2025, doi:10.3390/biomedicines13030566_

Round 1

Reviewer 1 Report

Comments and Suggestions for Authors

Dear Authors,

The article “ Insulin Like Growth Factor-I Reduces Collagen Production by Hepatic

Stellate Cells Through Stimulation of Collagen Degradation System via mTOR

Dependent Signaling Pathway” introduces a new knowledge about  IGF-I mechanism of action.

There are following remarks and comments.

1. Limited sample: The article does not specify the number of HSC cell lines used, which may affect the generalizability of the results. The use of only one clonal line may not reflect the heterogeneity of HSC in different liver conditions. Unclear experimental conditions: Cell culture conditions, including medium composition and incubation time, need to be more clearly described.

2.Conflicting conclusions: The article claims that IGF-I decreases collagen production, but at the same time there is a stimulation of DNA synthesis in HSC cells. This creates a logical inconsistency: if IGF-I stimulates cell proliferation, how is this consistent with its role in collagen reduction?

3. Please, explain using a t-test to analyze differences between groups. May it be insufficient when the number of samples is small? It is recommended that more powerful statistical analysis methods such as ANOVA or regression analysis be used for more complex data.

4. Fig.2 note is not clear and should be structured. “D” is absent, for example. Fig.5D – there is no statistics.

5. What are the implications of the observed down-regulation of TIMP-1 and up-regulation of MMP-2 and MMP-9 in response to IGF-I treatment for the understanding of extracellular matrix remodeling in liver pathology?

6. In the context of the mTOR-dependent signaling pathway, how does rapamycin's inhibitory effect on certain phosphorylation events inform our understanding of IGF-I's mechanism of action in hepatic stellate cells? Please, add the discussion.

7. Given that IGF-I did not significantly alter TGF-beta levels or type I procollagen mRNA expression, what alternative pathways or factors might be influencing collagen dynamics in hepatic stellate cells that warrant further investigation?

Author Response

Dear Dr. reviewer

Thank you very much for your comment. I've added the following.

  1. We studied collagen production and expression of related factors utilizing a cHSC, which maintains their phenotype of moderately activated hepatic stellate cells during culture including expression of IGF-I receptor, to elucidate the role of hepatic stellate cells on hepatic fibrogenesis when IGF-I is administered to cirrhotic livers .

We added the method of culture about this cell.

  1. As mentioned in the discussion, this may be due to the heterogeneity of hepatic stellate cells. It is necessary to examine the side effects of the treatment and whether the method of administration should be improved.

  1. Because the variance was small, a t-test was used.

  1. We added 2D in the manuscript. Figure 5D shows significant downregulation of TIMP1 after IGF-I administration. Although the statistical content is not shown to avoid cluttering the image, there is a significant difference between the control and IGF-I administration at 0.1nM and 0.5nM. TIMP1 tends to be canceled of downgulation depending on the concentration of rapamycin administered, and there is no significant difference from the control at 2.5nM administration. This result is significant by Jonckheere-Terpstra test, one of nonparametric tests. I will send the PDF.

  1. IGF-I treatment may have an anti-fibrotic effect by enhancing the degradation system rather than the synthesis system.

  1. We added explanations.

  1. IGF-I treatment may have an anti-fibrotic effect by enhancing the degradation system rather than the synthesis system.

As mentioned in the discussion, this may be due to the heterogeneity of hepatic stellate cells. It is necessary to examine the side effects of the treatment and whether the method of administration should be improved.

Thank you very much for your confirmation.

Sincerely yours,

Takako Nishikawa, M.D., Ph. D.

Department of Gastroenterology, Graduate School of Medicine, The University of Tokyo

Reviewer 2 Report

Comments and Suggestions for Authors

In this study the authors showed that IGF-1 is beneficial in cirrhosis of the liver in an in vitro system by Istimulating collagen degradation system by HSCs through mTOR dependent pathway.  i

The major issue with this study is the absence of an in vivo verification of the results obtained.

Colchicine is used for the  treatment of cirrhosis of the liver. I am not sure whether a comparison is possible between IGF-1 and colchicine by the authors. 

Insulin has anti-inflammatory actions and it will be interesting if the authors can see whether IGF-1 also has anti-inflammatory actions. Cirrhosis is an inflammatory event. Hence, I suggest that authors look at he plasma and hepatic concentrations of IL-6, TNF and see whether IGF- can modulate their actions and compare with those of colchicine.

Author Response

Dear Dr. reviewer

Thank you very much for your comment. I've added the following.

Tank you for pointing that out.

I will consider comparing it with colchicine. Would the drug you mentioned be colchicine, not somatomedin?

We used the rat hepatic stellate cell line, CFSC-8B. These cells have a small amount of lipid droplets containing vitamin A. They also have platelet-derived growth factor (PDGF-β) receptors and respond to PDGF. They express TGF-β, and although they do not express type IV collagen, they do express type I and III collagen and fibronectin. They do not express IL-6, which is seen in portal vein fibroblasts. In other words, we conducted our research assuming that these are moderately activated hepatic stellate cells.

We studied collagen production and expression of related factors utilizing a cHSC, which maintains their phenotype of moderately activated hepatic stellate cells during culture including expression of IGF-I receptor, to elucidate the role of hepatic stellate cells on hepatic fibrogenesis when IGF-I is administered to cirrhotic livers . As stated in the conclusion, further precise in vivo experiments would be required to clarify clinical relevance of IGF-I administration in the patients.

Thank you very much for your confirmation.

Sincerely yours,

Takako Nishikawa, M.D., Ph. D.

Department of Gastroenterology, Graduate School of Medicine, The University of Tokyo

Round 2

Reviewer 1 Report

Comments and Suggestions for Authors

Dear Authors, the article has been significantly improved.

Author Response

Dear Dr. reviewer

Thank you very much for your confirmation.

Sincerely yours,

Takako Nishikawa, M.D., Ph. D.

Department of Gastroenterology, Graduate School of Medicine, The University of Tokyo

Reviewer 2 Report

Comments and Suggestions for Authors

the study is well done, and results are appropriate and discussion is reasonable. 

Author Response

(The authors gave the same response as above.)
